# Phytase-Producing *Rahnella aquatilis* JZ-GX1 Promotes Seed Germination and Growth in Corn (*Zea mays* L.)

**DOI:** 10.3390/microorganisms9081647

**Published:** 2021-07-31

**Authors:** Gui-E Li, Wei-Liang Kong, Xiao-Qin Wu, Shi-Bo Ma

**Affiliations:** 1Co-Innovation Center for Sustainable Forestry in Southern China, College of Forestry, Nanjing Forestry University, Nanjing 210037, China; liguie2525@163.com (G.-E.L.); k3170100077@njfu.edu.cn (W.-L.K.); msb_817@126.com (S.-B.M.); 2Nanning Golden Camellia Park, Nanning 530022, China; 3Forestry Administration of Bazhong Municipality, Bazhong 636000, China

**Keywords:** *Rahnella aquatilis*, indole acetic acid, phytase-related soil properties, phytase activity, growth-promotion

## Abstract

Phytase plays an important role in crop seed germination and plant growth. In order to fully understand the plant growth-promoting mechanism by *Rahnella* *aquatilis* JZ-GX1, the effect of this strain on germination of maize seeds was determined in vitro, and the colonization of maize root by *R. aquatilis* JZ-GX1 was observed by scanning electron microscope. Different inoculum concentrations and Phytate-related soil properties were applied to investigate the effect of *R. aquatilis* JZ-GX1 on the growth of maize seedlings. The results showed that *R. aquatilis* JZ-GX1 could effectively secrete indole acetic acid and had significantly promoted seed germination and root length of maize. A large number of *R. aquatilis* JZ-GX1 cells colonized on the root surface, root hair and the root interior of maize. When the inoculation concentration was 10^7^ cfu/mL and the insoluble organophosphorus compound phytate existed in the soil, the net photosynthetic rate, chlorophyll content, phytase activity secreted by roots, total phosphorus concentration and biomass accumulation of maize seedlings were the highest. In contrast, no significant effect of inoculation was found when the total P content was low or when inorganic P was sufficient in the soil. *R. aquatilis* JZ-GX1 promotes the growth of maize directly by secreting IAA and indirectly by secreting phytase. This work provides beneficial information for the development and application of *R. aquatilis* JZ-GX1 as a microbial fertilizer in the future.

## 1. Introduction

Plant growth-promoting rhizobacteria (PGPR) are generally defined as a heterogeneous group of beneficial bacteria that are free-living in the plant rhizosphere and capable of promoting plant growth [1,2]. PGPR have been the subject of extensive research because of their profound effects on plant growth by direct and indirect modes of action [3,4,5]. The production of phytohormones, particularly indole acetic acid (IAA), auxins, gibberellins and cytokinins, are commonly referred to as a direct growth-promoting mechanism for PGPR [6,7,8]. In addition to the production of phytohormones, the mobilization of phosphorus (P) by PGPR is also one of the most studied aspects [9,10,11].

Phytate is one of the most abundant organic molecules containing unavailable P in soil. Phytate can account for 20% to 60% of soil organic P and can contain an amount of P that is equivalent to two-thirds of the P applied in fertilizer each year [12]. Plants depend on microorganisms to degrade phytate, and microbial phytases have been suggested to be useful in mineralizing P from phytate in soil and making P available for plant uptake [13,14]. In recent decades, many phytase-producing PGPR have been reported to promote plant growth [15,16,17,18].

The inconsistency between the laboratory and field effect are a major limitation on using plant growth-promoting rhizobacteria (PGPR) in agriculture because the soil is complex and involves a high number of factors. Most of the experiments have been carried out in greenhouse, but plant responses are less consistent and mainly dependent on soil type when experiments are conducted using soils [19]. A considerable amount of information suggests that P-related soil properties such as total P and phytate content, phosphorus-fixation capacity and pH have a major influence on plant responses to PGPR [20,21]. Therefore, the plant response to inoculation with phytase-producing PGPR is modulated by soil properties, especially those related to P. In addition, other factors, e.g., bacterial phytohormone production and concentration-dependent effects, may influence plant responses at the same time [22,23,24]. Thus, understanding the effect of these factors on PGPR performance is necessary in the process of implementing PGPR technology.

In China, the planting area of corn is approximately 24 million km^2^, is distributed in approximately 24 provinces and ranks second in the world [25]. With the development of animal husbandry and the corn processing industry, corn will become increasingly important in grain production in China. The high cropping index, excessive reclamation planting, short harvest period and excessive application of synthetic P fertilizers that occur in many corn planting areas in China will have destructive effects on soil ecological conditions, cause a decline in soil fertility and, subsequently, detrimentally affect corn growth.

*R. aquatilis* JZ-GX1 was isolated from pine rhizosphere soil and shows high phytase activity, which can promote the growth of poplar and Masson pine [26]. However, in addition to promoting forest tree growth, whether it can promote the growth of crops or whether there are other growth-promoting substances remain unclear. The aims of the study were to (1) investigate the growth-promoting effects of *R. aquatilis* JZ-GX1 on corn, including its influence on corn seed germination, its colonization of seedling roots, and the effects of soil P-related properties and inoculum concentration on seedling growth promotion; (2) discuss the growth promotion mechanism of *R. aquatilis* JZ-GX1; and (3) provide an efficient bacterial resource for biofertilizer use in agricultural production in the future.

## 2. Materials and Methods

### 2.1. Bacterial Inoculum Preparation

*R. aquatilis* JZ-GX1 was isolated from Masson pine rhizosphere soil, it was deposited in the China Center for Type Culture Collection (CCTCC, NO: M 2012439) [26]. Single colonies were sampled from nutrient agar plates and added to 50 mL fresh nutrient broth medium. The mixture was shaken at 180 rpm and 28 °C for 24 h, and then centrifuged at 10,000× *g* and 4 °C for 10 min. The pellet was then resuspended in physiological saline solution (9 g NaCl/L) to maintain osmotic balance. Then, the supernatant was adjusted to three concentrations, 10^5^, 10^7^ and 10^9^ cfu/mL for use.

### 2.2. Seed Germination Assay

The corn (*Zea mays* L.) seeds used in this study were obtained from the Corn Research Center of Beijing Agriculture and Forestry Academy, China. The seed germination experiment was carried out as described by Tiwari et al. [27] with a minor modification. The corn seeds were surface-sterilized by immersing in 70% ethanol for three minutes followed by rinsing three times with sterile distilled water (SDW), submerging in 10% (*v*/*v*) sodium hypochlorite for 15 min and subsequent washing 5 times with SDW. The seeds were imbibed in each of the three suspensions of the JZ-GX1 concentrations of 10^5^, 10^7^ and 10^9^ cfu/mL, respectively for 10 min and air-dried. Seeds soaked in sterile water alone served as the control (CK). Seeds placed in Petri plates (15 cm in diam) containing 0.7% water agar were incubated at 25 °C in the dark. Each treatment had three replicates, and each replicate contained 50 corn seeds. The germinated seeds were counted after 5 days of incubation, and the germination rate was computed. The formula of calculation was as follows: germination percentage (%) = (No. of seeds germinated/No. of seeds sown) ×100%. Ten seedlings were randomly chosen from each treatment for seedling length measurements (from the tip of the primary root to the apex of the shoot). Vigour Index = Germination percentage × Seedling length.

### 2.3. Quantitative Analysis of Indole Acetic Acid

The capacity of *R. aquatilis* JZ-GX1 to produce indole acetic acid (IAA) was determined in vitro. The strain was incubated for 24 h in Tryptic Soy Broth (TSB) medium, and 100 μL aliquots were transferred into 100 mL flasks containing 50 mL of TSB medium supplemented to reach 0, 100, 200, 300, and 500 μg/mL of L-tryptophan. One litre of TSB medium containing 15 g tryptone, 5 g soya peptone, 5.0 g NaCl, and 1000 mL distilled water, and 0.1 M NaOH solution was used to adjust the pH to 7.2 ± 0.2. L-tryptophan was added as a filtered sterilized (0.22 μm) 2 mg/mL stock solution prepared with warm sterile distilled water (with three replicates of each L-tryptophan concentration) [28]. The flasks were placed in an orbital shaker (180 rpm) at 28 °C. Samples were taken after 24, 48, and 72 h inoculation. An aliquot of the suspension in each flask was centrifuged (10,000× *g*, 15 min) to remove bacterial cells. A total of 1 mL of supernatant was mixed with 4 mL of Salkowski’s reagent (150 mL 18 M H_2_SO_4_, 250 mL distilled water, 7.5 mL 0.5 M FeCl_3_·6H_2_O) and measured at an absorbance of 535 nm after incubation for 30 min at 40 °C [29,30]. The IAA concentration was determined by comparison with a standard curve prepared with an indole acetic acid standard sample (Sigma I-2886).

### 2.4. Colonization of Corn by JZ-GX1 In Vitro

Seed disinfection and germination were performed as described above for the seed germination assay, and then 2-day-old seedlings with roots of approximately 2 cm were selected for inoculation, dipped into the culture (10^7^ cfu/mL) and gently swirled for 10 min. Finally, the inoculated seedlings were grown in sterilized basal Murashige-Skoog medium containing 0.8% agar (without sucrose) and incubated in a versatile environmental test chamber (25 °C, 16 h light, 8 h darkness).

The roots of the corn were sampled and scanned under electron microscopy 15 days after planting [31]. A 1 cm primary root segment was taken from each part of the root of the corn seedlings. The segment was also gently rinsed with sterile distilled water and then divided into two 5 mm segments and processed for scanning electron microscopy. The segments were fixed with glutaraldehyde. After rinsing several times in Na-cacodylate buffer solution, specimens were post-fixed for 4.5 h in 1% osmium tetroxide at 4 °C and washed again in Na-cacodylate buffer solution. The root segments were dehydrated by a graded series of ethanol solutions and 100% acetone and then dried by critical point drying with liquid CO_2_ (EMITECH K850). The samples were subsequently mounted on stubs, sputtered with gold (HITACHI E-1010) and examined using a scanning electron microscope (QUANTA 200, FEI, Chicago, IL, USA).

### 2.5. Soil-Plant Experiment

Soil from a mountain near Nanjing Forestry University, China, was selected because of its naturally low total P content and low level of organic matter. A routine soil analysis was performed in the Soil Testing Laboratory at Nanjing Forestry University. The soil pH was 5.1, organic matter 1.5%, and nutrient content (mg/kg): total P 3.12, effective P 0.39, K 499.92, 144 Ca 202.62, Mg 249.9, Fe 1410.6, Mn 30.66, Zn 0.24, and Cu 2.22. These routine soil analysis was performed in the Soil Testing Laboratory at Nanjing Forestry University.

Seed disinfection and germination were performed as described above. Afterwards, 2-day-old seedlings with roots of approximately 2 cm in length were planted individually in plant growth containers that contained of 400 g of a mixture of 1:1 (*w*/*w*) soil–sand (autoclaved at 121 °C for 90 min to eliminate the native microflora). The amount of soil (air-dried and sieved at <4 mm) and sand (200 g each) to be added into each container were weighed separately and then mixed to ensure homogeneity. The experiment was conducted following a 4 × 4 factorial design. Four different P regimes were evaluated: no P addition, 20 mg Pi kg^−1^ soil, 95.5 mg phytate kg^−1^ soil (equivalent to 20 mg Pi kg^−1^ soil, 1×), and 286.6 mg phytate kg^−1^ soil (equivalent to 60 mg Pi kg^−1^ soil, 3×) [32,33]. There were four bacterial treatments: no inoculation (only SDW) and three concentrations of JZ-GX1, 10^5^, 10^7^ and 10^9^ cfu/mL. Ten millilitres of the bacterial suspensions at each concentration were inoculated around seedling roots. Ten 2-day-old seedlings planted in containers were used for each combination treatment. The soil moisture was kept between 40–60% of the maximum water holding capacity, and plants were grown at 25 °C in a growth chamber with 16 h light and 8 h dark. After 30 days, the various parameters were determined as described below.

### 2.6. Determination of Photosynthetic Parameters

The gas exchange parameters of the third or fourth fully expanded mature leaves below the terminal bud, including the net photosynthetic rate (Pn, μmol CO_2_ m^−^^2^ s^−1^), stomatal conductance (gs, mmol H_2_O m^−2^ s^−1^), intercellular CO_2_ concentration (Ci, μmolCO_2_ mol air^−1^) and transpiration rates (E, mmol H_2_O m^−2^ s^−1^), were recorded using a portable photosynthesis system (LI-6400XTR; Li-Cor Inc., Lincoln, NE, USA). All measurements were carried out in a growth chamber under an artificial light source with a photosynthesis photon flux density (PPFD) of 800 μmol m^−^^2^ s^−1^ provided by an LI-6400 LED light source with a vapor pressure deficit of 2.12 ± 0.06 kPa and an ambient CO_2_ concentration of 380 ± 5 μmol CO_2_ mol air^−1^. There were five replicates per treatment combination (one leaf per replicate).

### 2.7. Determination of Chlorophyll Content

The pigments were extracted according to the method of Tian et al. [34] with some modifications. From the third or fourth fully expanded fresh leaves below the terminal bud, 0.2 g of the leaf blade was cut out, sliced into 1–2 mm wide strips and immersed into 10 mL of acetone-alcohol extract (*v*/*v*,1:1) until all the strips turned completely white. Chlorophyll a + b contents were determined spectrophotometrically with a Thermo Spectronic HEλIOS γ (Waltham, MA, USA) at 663 and 646 nm, respectively. The pigment contents were calculated according to Lichtenthaler et al. [35]. Each treatment combination had five replicates (one leaf per replicate).

### 2.8. Measurement of Phytase Activity in the Root

Root phytase activity (myoinositol 2-monophosphate) was determined following the methods described by Quan et al. [36] with a minor modification. The roots of the corn plants were rinsed in deionized water, and 0.5 g roots were incubated for 90 min in 10 mL exudation solution containing a 5 mM maleate buffer with pH 5.5, 2% sucrose, 2 mM CaCl_2_ and 0.01% (*v*/*v*) protease inhibitor cocktail (Sigma). The exudates were then measured for phytase activity. For phytase activity measurements, 1 mL acetate buffer (0.2 M, pH 5.5 containing 1 mM sodium phytate) and 2 mL of exudation solution (2 mL of SDW was used as the control treatment) were incubated for 30 min at 37 °C. Then, the reaction was terminated by adding 1 mL of 10% tricholoroacetic acid (TCA). Free phosphate was determined by a modified colorimetric molybdate blue method [37]. One unit (U) of phytase activity was defined as the amount of enzyme releasing 1 μmol of Pi equivalent per minute. Each treatment combination had five replicates.

### 2.9. Measurement of Plant Total P Concentration

After incineration at 550 °C and solubilization of the ash in 6 M HCl, the concentrations of P in plants were calculated using ICP emission-spectroscopy (ICP-AES Optima 2100DV, Perkin Elmer, MA, USA) according to standard procedures. There were five replicates for each treatment combination.

### 2.10. Biomass Determination

Corn plants harvested from the above experiments were washed with distilledwater and blotted dry. The fresh weight (FW) of the plants was recorded. Plants were divided into roots and shoots and dried in a hot-air oven at 80 °C. Each treatment combination had five replicates. The dry weight was recorded every 12 h until reaching a constant weight.

### 2.11. Statistical Analysis

Data analysis was carried out using SPSS Statistics 18.0 (IBM Inc., Armonk, NY, USA), charts making used Excel (Microsoft Office Excel 2007). When the experiments considered two factors (Inoculation concentration and phytase content) two-way ANOVA was used.

## 3. Results

### 3.1. Effects of R. aquatilis JZ-GX1 on Seed Germination and Root Elongation in Corn

The seed germination results in the different bacterial concentration treatments are given in Figure 1. The results showed that *R. aquatilis* JZ-GX1 significantly promoted seed germination. On the fifth day after sowing, the 10^5^ cfu/mL and 10^7^ cfu/mL *R. aquatilis* JZ-GX1 treatments performed significantly better than other treatments. The highest germination rate (94.12%), seedling length (9.8 cm), and seed vigour index (924.59) were obtained at the inoculum concentration 10^7^ cfu/mL, whereas the highest inoculum concentration (10^9^ cfu/mL) showed no effect or lower values than the non-inoculated control.

### 3.2. Indole Acetic Acid Produced by R. aquatilis JZ-GX1

The soil-independent root growth-promoting effect observed in the seedlings after inoculation with the JZ-GX1 bacterial suspension suggests the involvement of a hormone-like compound, perhaps an auxin-type compound. To further clarify this, we tested whether *R. aquatilis* JZ-GX1 could produce indole acetic acid (IAA), a widely known bacterially produced auxin. The results are shown in Figure 2. This strain was able to effectively produce IAA in vitro. The production of IAA was additionally stimulated by the presence of tryptophan in the growth medium. Even in the absence of tryptophan, a small amount of IAA was detected in the medium after 24 h. The production of this compound increased significantly with the increase of tryptophan and time. When 500 μg tryptophan mL^−1^ was added, the highest yield of IAA in the tests, 10.5 μg/mL, was achieved after 72 h incubation.

### 3.3. Colonization of Corn Seedlings by R. aquatilis JZ-GX1 In Vitro

The observation using Scanning Electron microscope (SEM) revealed that *R. aquatilis* JZ-GX1 colonized the corn root well. On the highly colonized sites, such as the concavities of root surfaces, a number of JZ-GX1 cells were easily observed (Figure 3B,C). Root hairs and the surrounding epidermis were also observed to be popular habitats for JZ-GX1 (Figure 3D). Thus, it was proven that JZ-GX1 is endophytic, at least in corn.

### 3.4. Effects of R. aquatilis JZ-GX1 on Seedlings in the Soil-Plant Experiment

When the seedlings were grown under growth chamber conditions for 30 days, the P regime, bacterial inoculation and their interaction had highly significant effects on the Pn, gs, E, Ci, Chl (a + b), root phytase activity, total P concentration, and biomass of corn seedlings. Specifically, bacterial inoculation led to a significant increase in those response variables evaluated only in combinations with phytate (1× and 3×, which corresponded to 95.5 and 286.6 mg phytate kg^−1^ soil, equivalent to 20 and 60 mg Pi kg^−1^ soil). The difference was more significant at the higher rate of phytate, whereas a weaker effect was found under the other two P regimes. In the treatment, the combination of 3× phytate with three different concentrations of *R. aquatilis* JZ-GX1 culture caused highly significant differences from the non-inoculated control.

### 3.5. Photosynthetic Parameters and Chlorophyll Content of Corn Seedlings

Compared to those of the treatment with no addition of P, the Pn, gs and E of the three other treatments (1×, 3×, Pi) increased significantly, but Ci was reduced to different degrees (Figure 4A–C). Furthermore, under the treatments with different phytate contents, there were significant differences in photosynthetic parameters between the inoculated treatments and the non-inoculated treatments. The effect of the 3× phytate treatment was the most significant of the four different P regimes. The results showed that different concentrations of phosphorus improved the photosynthetic capacity of corn. The net photosynthetic rate in corn seedlings at the light saturation level significantly increased at the higher rate of phytate when inoculated with JZ-GX1, which was attributable to an increase in stomatal conductance, which then accelerated the transpiration rate. The intercellular CO_2_ concentration and net photosynthetic rate were negatively correlated; the higher the transpiration rate was, the lower the intercellular CO_2_ concentration (Figure 4D). The information in Figure 4A and Figure 5C shows consistency in the net photosynthetic rate and chlorophyll content. The total chlorophyll content in the treatment with 3× phytate and 10^7^ cfu/mL bacterial suspension was the highest of those in all the treatments; its net photosynthetic rate was also the highest.

### 3.6. Phytase Activity in the Root and Total P Concentration of Plants

All plants in the treatments that added phytate into the soil showed increased phytase activity (3–7 times) compared to control plants (Figure 6A). Although all inoculation rates resulted in significant increases in comparison with the non-inoculated control when phytate was present, the phytase activity in the root of plants in the treatment with 3× phytate and 10^7^ cfu/mL bacterial suspension was higher than that in the other treatments, up to 8.17 mU/g. For the total P content of the plants, among all the treatments, the highest total P content was also observed in the treatment with 3× phytate and 10^7^ cfu/mL bacterial suspension (Figure 6B). The total P content in that treatment was 7.35 times greater than that in the control. The enhanced P uptake in the plants can be explained by the mobilization of P from the added phytate under the effect of the higher rhizosphere phytase activity. The results indicated that the secretion of phytase from the corn roots enabled plants to take up additional P from soil with poor soluble P that was supplemented with insolvable phytate.

### 3.7. Effect of R. aquatilis JZ-GX1 on Biomass Accumulation in Corn Seedlings

Physiological change is a gradual process. In the end, the results are shown in the changes in plant biomass (fresh weight and dry weight). The biomass accumulation of corn seedlings inoculated with different concentrations of bacterial suspensions of JZ-GX1 and planted in soil supplemented with Pi or phytate for 30 days is depicted in Figure 5A,B. The maximum increase in both fresh weight and dry weight was obtained from the treatment with 3× phytate and 10^7^ cfu/mL bacterial suspension, and the differences from the other treatments were significant. A decline in biomass was observed at the same inoculation concentration when the P regimes were changed to no P addition, 1× phytate and Pi addition. In the other treatment combinations, the biomass yields were slightly higher than that in the control.

## 4. Discussion

At present, many studies on plant growth-promoting rhizobacteria and their effects on plant growth have been reported. There are different approaches to using PGPR, including direct inoculation with PGPR [38,39,40], inoculation with transgenic microorganisms [41,42], and the use of transgenic plants that express microbial enzymes [43,44,45].

Whether PGPR play an important role in growth promotion mainly depends on the effective colonization of the plant roots after inoculation [46,47]. It is usually accepted that establishing efficient colonization on plant roots is a critical step for PGPR to proceed with plant-microbe interactions [48]. In our study, *R. aquatilis* JZ-GX1 cells successfully colonized the corn roots. This is consistent with results obtained for colonization by another kind of gram-negative *Pseudomonas* [49]. Areas of heavy JZ-GX1 colonization were usually found at concave parts of the epidermal surface, at sites where lateral roots appear, and on root hairs.

It has received wide attention that some PGPR can secrete phytohormones, and the production of IAA is the most studied case [50]. IAA accumulation in the rhizosphere has also been linked to the inoculum density of the applied PGPR strain [51]. In our research, this inference was verified. The phytase-producing strain *R. aquatilis* JZ-GX1 had a strong ability to synthesize IAA; even when no L-tryptophan was added, a small quantity of IAA was detected in the medium. Identifying an appropriate inoculation concentration in the gnotobiotic seed germination assay was challenging. When the inoculation concentration was too high, it inhibited the germination of the corn seeds. The possible reasons why germination is inhibited at high cells concentration, possible links with over-auxin production and other factors. The inoculation of corn seeds with JZ-GX1 at 10^7^ cfu/mL greatly increased seed germination, root elongation, and the vigour index of corn and led to highly significant results. Therefore, JZ-GX1 can directly affected plant growth, probably due to its IAA production.

In the soil-plant experiment, *R. aquatilis* JZ-GX1 significantly promoted plant growth and P uptake compared with those of the controls. However, these effects were only observed when phytate was added to the low-P soil and the strain JZ-GX1 was inoculated into the corn seedlings. The most beneficial effect on plant growth came from the combination of 3× phytate and 10^7^ cfu/mL JZ-GX1 suspension, which promoted plant growth significantly more than the other treatments. It was found that the 10^7^ cfu/mL JZ-GX1 suspension was the optimal inoculation concentration for corn, both in the seed germination assay and in the soil-plant experiment. This was suggested by the concentration-dependent effect of inoculation observed in the plant soil experiment. This is consistent with the studies of Kammoun et al. [52] and Bhakta et al. [53].

Under experimental conditions, *R. aquatilis* JZ-GX1 only significantly increased the growth of corn seedlings when phytate was added to the soil, i.e., under circumstances where organophosphate levels are sufficient, but the P is fixed in the soil. No significant effect of inoculation was observed without phytate addition either under P-limited conditions (no P, low total P content) or under full inorganic P fertilization. The effects of inoculation with JZ-GX1 on corn seedlings only occurred under conditions that were conducive to phytase activity, which supported the previous supposition that phytase activity is the major mechanism of plant growth promotion by *R. aquatilis* JZ-GX1. It is important to mention that in natural soil systems, phytates are insoluble and not taken up by chelation with Ca^2+^, Fe^2+^, Mg^2+^, Zn^2+^, etc. [54]. Nevertheless, phytase activity degrades phytate to increase P uptake while also increasing the availability of phytate-chelated nutrients to promote plant growth [24]. This effect being produced by *R. aquatilis* JZ-GX1, a kind of phytase-producing PGPR, cannot be ruled out. Additionally, this strain produces IAA, which can create a concentration-dependent response to inoculation and interacts with phytase-mediated effects.

It was confirmed in a different experiment that the phytase secreted by *R. aquatilis* JZ-GX1 is an extracellular enzyme [26]. Phytase activity in the root was detected, and phytase activity was remarkably enhanced under conditions in which phosphorus was deficient and phytate and JZ-GX1 inoculum were added. Research results showed that JZ-GX1 strain can colonize on the surface and interior of maize’s roots, and therefore, phytases are present in roots. Phytases accumulated in the tissues as result of microbial activity, flow out of the roots after they have been immersed into a buffer. Although, phytases were in the roots, they were bacterial origin. The cells of this endophytic bacterium secreted phytases, and after that this enzyme could be slowly translocated into the rhizoplane and the follow into rhizosphere. However, very low phytase activity was detected in the treatments inoculated with the JZ-GX1 gradient bacterial suspension but without P or Pi addition. This result indicated that the phytase secreted by the root was an induced enzyme and that phytase synthesis could be induced to significantly increase under the conditions of phosphorus starvation and the presence of phytate in the soil.

Soil is an important factor that cannot be neglected in soil-plant experiments on the inoculation of P-solubilizing bacteria to promote plant growth. The composition of agricultural soils is complicated, which causes field inoculation applications and laboratory tests to have different results or even no effect. First, soil influences the physiological functions of the bacterial inoculation [22]. In addition, different soil factors, such as the contents of C and P in soil, also affect bacterial phytase production [55]. Moreover, soil properties interact with phytate (the substrate), phytase (the enzyme) and phosphorus (the product) [56]. In this research, the significant effects of phytase activity and the plant growth promotion activity of JZ-GX1 were only observed when phytate existed in the soil at levels equivalent to having enough organophosphates but was fixed in the soil. No effect of JZ-GX1 inoculation occurred when the total P content was low or when inorganic phosphorus was sufficient in the soil.

The use of PGPR as bacterial biofertilizer to increase crop yield in agriculture faces several challenges. To achieve this goal, we need to understand the interaction between plants and bacteria and to grasp the environmental factors that affect their interaction. The results of the present study showed that the soil adjusted the reaction between plants and bacteria in a particular way. In the field application of phytase-producing rhizobacteria, some variables are difficult to control. Therefore, to maximize the plant growth promotion effect, the total P and phytate contents of the soil have to be determined, and the survival rate of the inoculum needs to be evaluated. Clearly understanding the mechanisms of action of PGPR and their interactions with plants and soil is the only way to predict the conditions under which plant growth promotion will occur. Hence, our research will be an important step in improving the consistency of growth promotion by PGPR.

## Figures and Tables

**Figure 1 microorganisms-09-01647-f001:**
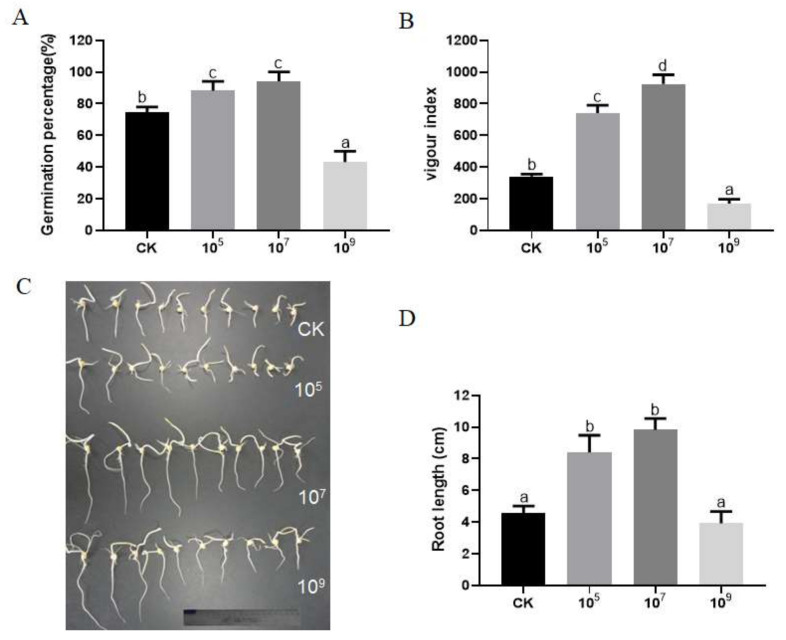
The seed germination percentage (**A**), vigour index (**B**) and root length (**C**,**D**) of corn under four concentrations of the *R. aquatilis* JZ-GX1. Vertical SD bars were added above the histograms, and the different letter above the SD bars indicates very significant difference according to Tukey’s HSD at *p* ≤ 0.01. The portion of a ruler painted in blue in panel C is 2 cm.

**Figure 2 microorganisms-09-01647-f002:**
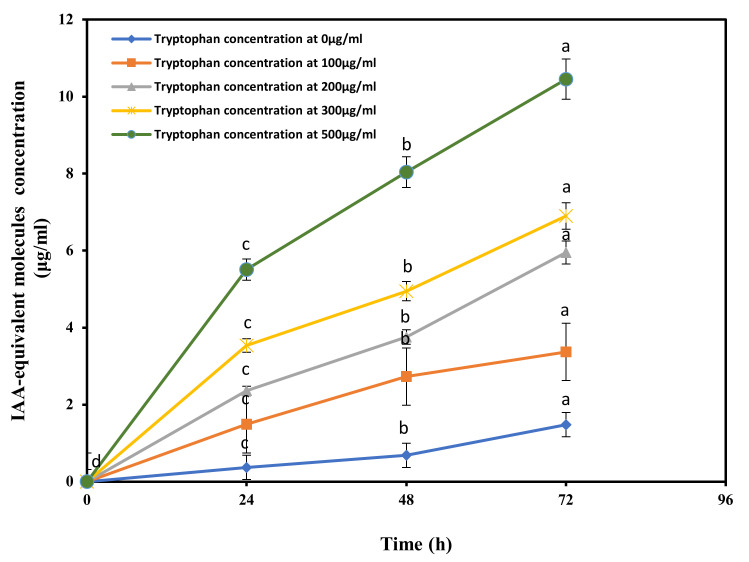
Production of indole acetic acid (IAA) by *R. aquatilis* JZ-GX1 in the presence of L-tryptophan at various concentrations. Vertical SD bars were added above the line charts, and the different letter above the SD bars indicates very significant difference according to Tukey’s HSD at *p* ≤ 0.01.

**Figure 3 microorganisms-09-01647-f003:**
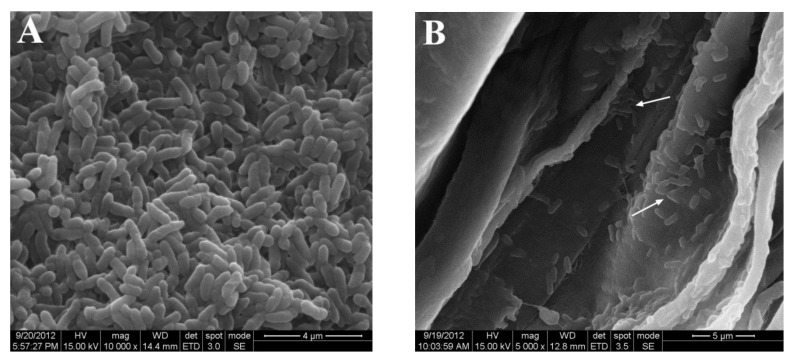
Scanning microscopy images of *R. aquatilis* JZ-GX1 colonizing corn roots in a gnotobiotic system. (**A**). Bacterial cells of JZ-GX1. (**B**,**C**). JZ-GX1 cells on the concave parts of root surfaces. (**D**). JZ-GX1 cells on a root hair and the surrounding epidermis. The white arrows indicate the JZ-GX1 cells in all panels. The black arrow points to the root hair in D.

**Figure 4 microorganisms-09-01647-f004:**
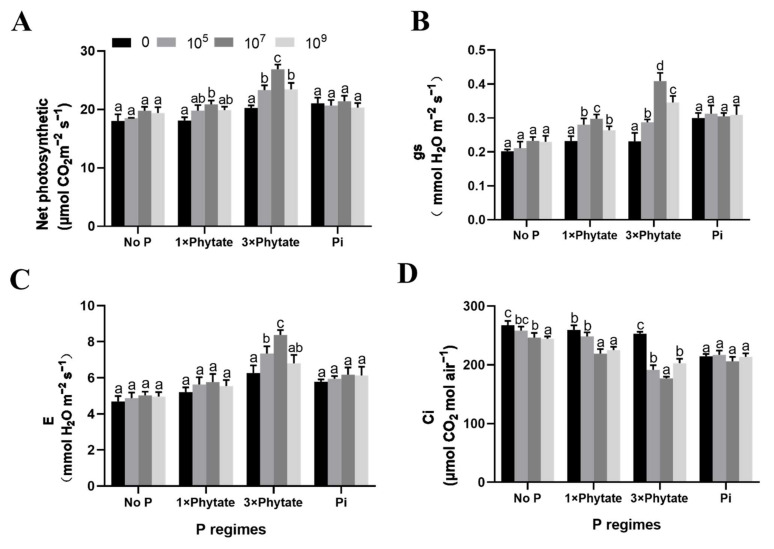
Effect of inoculation with three different concentrations of *R. aquatilis* JZ-GX1 under four different P regimes on the photosynthetic parameters of corn seedlings. (**A**) Net photosynthetic rate (Pn), (**B**) Stomatal conductance (gs), (**C**) Transpiration rate (E), (**D**) Intercellular CO_2_ concentration (Ci). Different letters indicate statistically significant differences (*p* < 0.05) among treatments according to the least significant difference test.

**Figure 5 microorganisms-09-01647-f005:**
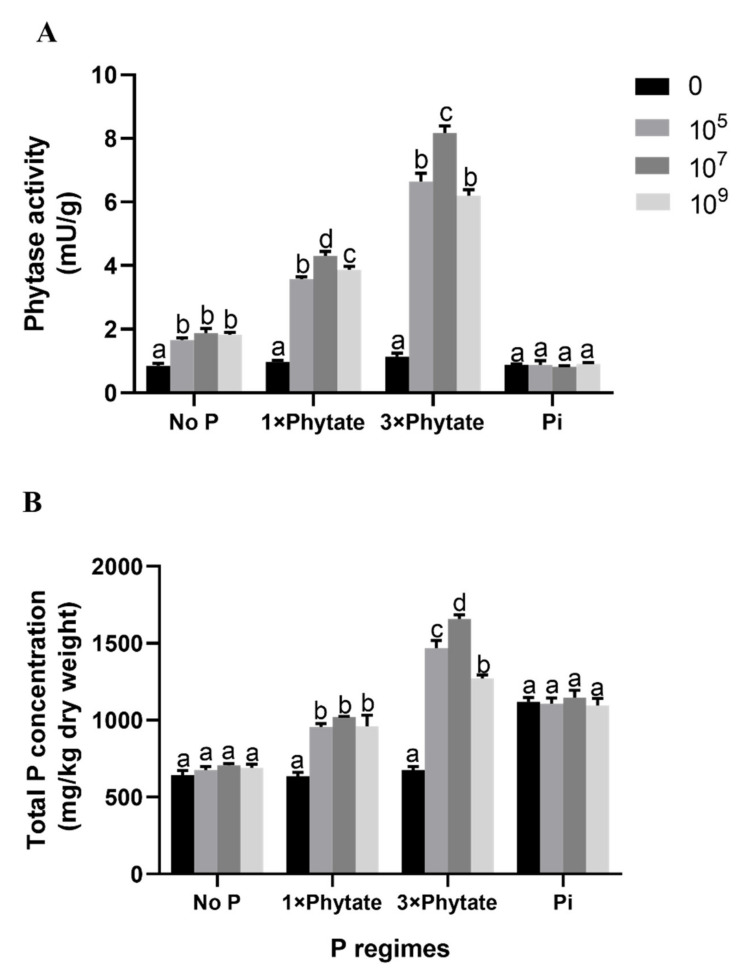
Phytase activity (**A**) in exudates from roots in the rhizosphere and total P concentrations (**B**) of corn plants inoculated with three different concentrations of *R. aquatilis* JZ-GX1 under four different P regimes in a soil. Different letters indicate statistically significant differences (*p* < 0.05) among treatments according to the least significant difference test.

**Figure 6 microorganisms-09-01647-f006:**
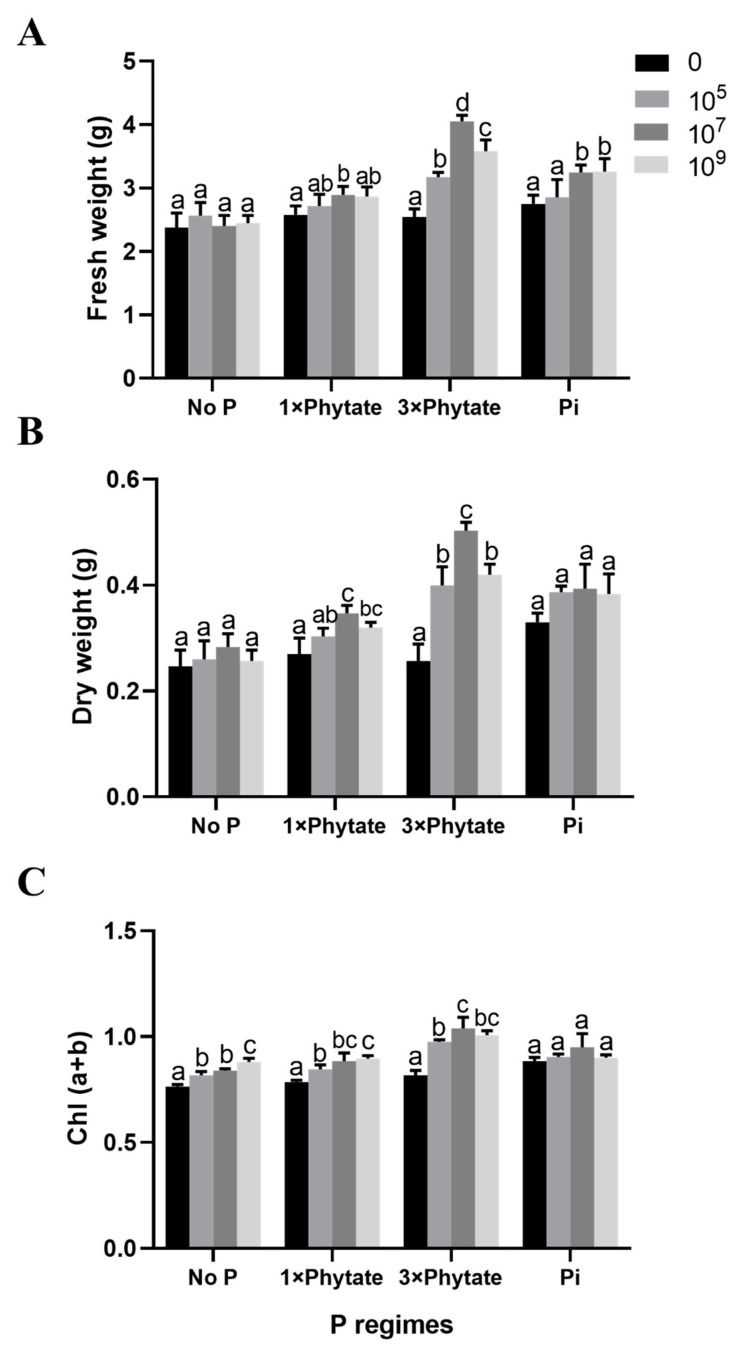
Effects of inoculation with three different concentrations of *R. aquatilis* JZ-GX1 under four different P regimes on (**A**) Fresh weight accumulation, (**B**) Dry weight accumulation and (**C**) Chlorophyll content. Different letters indicate statistically significant differences (*p* < 0.05) among treatments according to the least significant difference test.

## Data Availability

All the data and materials have been provided in main manuscript.

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
