# Peer review of "Phytase-Producing Rahnella aquatilis JZ-GX1 Promotes Seed Germination and Growth in Corn (Zea mays L.)"

_microorganisms, 2021, doi:10.3390/microorganisms9081647_

Round 1

Reviewer 1 Report

Dear authors,

The present work shows an interesting investigation about the use of the JZ-GX1 bacterium as a bioinoculant in corn crops, relating its use with the use and selective production of phytases under conditions of the presence of the substrate in the soil. A priori, it is a complete investigation and carried out at levels from laboratory to plant assays.

However, there are a series of questions that must be answered or solved in the manuscript in order to determine if the article can be published.

-Please include citation 26 in materials and methods to clearly indicate the provenance of the strain and the attributed characteristics in addition to the introduction. Furthermore, as a suggestion an update of the identification of this strain could be included, since at present this strain is not Ranhella aquatilis but R. woolbedingensis according to the sequence deposited in GenBank (KC351183.1). I recommend that the authors compare this sequence with the deposited database to observe the similarities with the species of the genus Rahnella described and not assign this strain to the wrong species.

-line 57: could the unit km2 be used instead of hm2 to express the area planted with corn in a more intuitive way?

-The determination of IAA production is usually carried out in minimal media or at least free of complex nitrogen sources that can interfere with the results obtained, including the incorporation of an unknown amount of tryptophan. What is the reason to use TSB instead of another medium without a complex nitrogen source such as tryptone and soy peptone that can have considerable concentrations of tryptophan?

-The analysis of colonization is an important point in the study of the PGP capacities of bacteria. In this regard, have they considered the use of the transconjugation technique to incorporate a plasmid that codes for a GFP and thus be able to observe the bacteria in vivo instead of the use of SEM that limits the observation of the distribution of bacteria in the root?

-A noteworthy aspect is the use of the concept of "root secreted phytases" when we are considering that it is the bacteria that produces them. Are these phytases produced by the bacteria or by the plant? Please clarify this matter.

-The concept "biobacterial manure" must be explained. Are there any bacteria that are not "bio"? perhaps the appropriate concept would be "bacterial biofertilizer" or "bacterial biostimulant" but manure refers to a type of product that does not correspond to a priori bacterial inoculant.

-line 52: change bold

-line 176, 186: change underline

Best regards

Author Response

Response to Reviewer 1 Comments

Point 1: Please include citation 26 in materials and methods to clearly indicate the provenance of the strain and the attributed characteristics in addition to the introduction. Furthermore, as a suggestion an update of the identification of this strain could be included, since at present this strain is not Ranhella aquatilis but R. woolbedingensis according to the sequence deposited in GenBank (KC351183.1). I recommend that the authors compare this sequence with the deposited database to observe the similarities with the species of the genus Rahnella described and not assign this strain to the wrong species

Response 1: According to the recommendations of the reviewers, we have cited reference [26] to clarify the source and attribution characteristics of the strains used, the details can be found in Line 77, Page 2. We re-compared the 16S rRNA sequence of JZ-GX1 strain in the NCBI database, and still found that it had the highest similarity with Rahnella, and we had sequenced the whole genome of this strain in the later stage, which showed the highest matching degree with Rahnella aquatilis, so there was no problem in the identification of this strain.

Point 2:line 57: could the unit km2 be used instead of hm2 to express the area planted with corn in a more intuitive way?

Response 2: Thanks to the reviewer for your suggestions, we have corrected the unit and the details can be found in Line 77, Page 2.

Point 3:The determination of IAA production is usually carried out in minimal media or at least free of complex nitrogen sources that can interfere with the results obtained, including the incorporation of an unknown amount of tryptophan. What is the reason to use TSB instead of another medium without a complex nitrogen source such as tryptone and soy peptone that can have considerable concentrations of tryptophan?

Response 3: The use of TSB medium to detect the ability of microorganisms to produce IAA is internationally recognized, because tryptophan contained in tryptone is negligible (Liu Wan Hui et al. Indole-3-Acetic Acid in Burkholderia pyrrocinia JK-SH007: Enzymatic Identification of the Indole-3-Acetamide Synthesis Pathway[J]. Frontiers in Microbiology, 2019). In addition, if the organic nitrogen source is replaced by inorganic nitrogen source, it may be difficult for bacteria to grow. And the synthesis of IAA itself takes tryptophan as the precursor. Our experiments also show that the amount of IAA synthesized by JZ-GX1 increases with the increase of tryptophan concentration. In the interaction between plants and microorganisms, microorganisms living in the plant rhizosphere will use the amino acids in root exudates to synthesize auxin, thus providing plant growth.

Point 4: The analysis of colonization is an important point in the study of the PGP capacities of bacteria. In this regard, have they considered the use of the transconjugation technique to incorporate a plasmid that codes for a GFP and thus be able to observe the bacteria in vivo instead of the use of SEM that limits the observation of the distribution of bacteria in the root?

Response 4: The reviewer's suggestion is to the point. It is indeed convenient to use fluorescent labeled strains to study the colonization of PGPR, but the unclear genetic background of JZ-GX1 wild-type strains has caused great trouble to our transformation, so we are trying to overcome this difficulty.

Point 5: A noteworthy aspect is the use of the concept of "root secreted phytases" when we are considering that it is the bacteria that produces them. Are these phytases produced by the bacteria or by the plant? Please clarify this matter.

Response 5: According to the results of our colonization experiment, JZ-GX1 can colonize in the root surface and interior of maize plant, we want to know the ability of JZ-GX1 strain to secrete phytase in maize rhizosphere, so we measured the phytase activity of maize root system.

Point 6: The concept "biobacterial manure" must be explained. Are there any bacteria that are not "bio"? perhaps the appropriate concept would be "bacterial biofertilizer" or "bacterial biostimulant" but manure refers to a type of product that does not correspond to a priori bacterial inoculant.

Response 6: We are very sorry for our negligence. We have replaced this word and the details can be found in Line 394, Page 14.

Point 7: line 52: change bold

Response 7: After our careful examination, we found no bold in line 52 of the manuscript.

Point 8: line 176, 186: change underline

Response 8: We are very sorry for our negligence. The corresponding underline has been modified, the details can be found in Line 176 and 186, Page 4.

Reviewer 2 Report

According to me, the manuscript entitled “Phytase-producing Rahnella aquatilis JZ-GX1 promotes seed germination and growth in corn (Zea mays L.)” fit within the journal scope. However, more microbiological tests should be provided for discuss the plant growth promotion mechanisms by R. aquatilis JZ-GX1. Authors did not investigate the phytase activity of R. aquatilis JZ-GX1 under the experimental conditions (soil-plant experiment). In my opinion phytase extracted from rhizosphere soil should be also tested, because phosphorus mobilization processes take place in this sphere.

Studies concern changes of the biometrics and biochemistry of plants caused by inoculation of seeds with phytase-producing bacteria. The Authors analyzed plant phytases (2.8. Measurement of root-secreted phytase activity, 3.6. Root-secreted phytase activity and total P concentration of plants), while the discussion is based on the analysis of the influence of microbial phytases on plant parameters. In my opinion, in this manuscript the lack consistency between results and the discussion is occurred. For this reason it is hard to me to evaluate the correctness of interpretation of these studies. Maybe the methods needs to be changed.

My doubts are raised by the method of sterilization. Did one-time sterilization eliminate all spores of microorganisms present in the soil? On the other hand, biofertilizers are applied into the microbiologicallly active soil. What is the viability of R. aquatilis JZ-GX1 in the soil inhabited by the native community of microorganisms? What is the activity of R. aquatilis JZ-GX1 in the microbiologically active soil?

In the abstract is written “…colonized on the root surface, root hair and the root interior of maize.”, and in chapter 3.3 - “… JZ-GX1 is epiphytic, at least in corn.” What is truth?  The same form should be used: “indole acetic acid (IAA)” or “indoleacetic acid (IAA)” According to me, the text requires linguistic correction, e.g.:

Line 11: “the growth-promoting mechanism of Rahnella aquatilis JZ-GX1” change to “the plant growth-promoting mechanism by Rahnella aquatilis JZ-GX1”

Lines 12-13: “the colonization of R. aquatilis JZGX1 in maize root” change to “the colonization of maize root by R. aquatilis JZGX1”

Line 65: “high phytase” change to „ high phytase activity”

Line 80: “resuspended in physiological” change to „resuspended in physiological saline solution”

Line 142: “pH instrument” change to „pH-meters”

Line 147: “consisted of 400 g” change to “contained of 400 g”

Line 159: “the response variables were determined” change to “the various parameters were determined”

Otherwise: 

Line 186: “measured” - Why underlined?

Line 200: “distilledwater” - Why underlined?

All figures are correctly cited in the text. The figures are informative. Table 1 is redundant. In my opinion, marking the changes significance by using letters is sufficient.

In my opinion this manuscript cannot be publish in this form.

Author Response

Response to Reviewer 2 Comments

Point 1: According to me, the manuscript entitled “Phytase-producing Rahnella aquatilis JZ-GX1 promotes seed germination and growth in corn (Zea mays L.)” fit within the journal scope. However, more microbiological tests should be provided for discuss the plant growth promotion mechanisms by R. aquatilis JZ-GX1. Authors did not investigate the phytase activity of R. aquatilis JZ-GX1 under the experimental conditions (soil-plant experiment). In my opinion phytase extracted from rhizosphere soil should be also tested, because phosphorus mobilization processes take place in this sphere.

Response 1: The results of relevant studies on the ability of the strain to secrete phytase under laboratory culture conditions have been published, the purpose of this study is to explore whether the strain can also play a bioactive role in the real soil environment. As the reviewers said, phosphorus mobilization does occur in the soil, but according to our research results, JZ-GX1 strain can colonize on the surface and interior of maize roots. The strain first secretes phytase activity in plant roots, and then slowly secretes into the soil.

Point 2:Studies concern changes of the biometrics and biochemistry of plants caused by inoculation of seeds with phytase-producing bacteria. The Authors analyzed plant phytases (2.8. Measurement of root-secreted phytase activity, 3.6. Root-secreted phytase activity and total P concentration of plants), while the discussion is based on the analysis of the influence of microbial phytases on plant parameters. In my opinion, in this manuscript the lack consistency between results and the discussion is occurred. For this reason it is hard to me to evaluate the correctness of interpretation of these studies. Maybe the methods needs to be changed.

Response 2: The author discussed the purpose of this study, combined with several important research results, and discussed not only the plant parameters, but also the growth-promoting mechanism of JZ-GX1, especially the part of soil phytate degradation.

Point 3:My doubts are raised by the method of sterilization. Did one-time sterilization eliminate all spores of microorganisms present in the soil? On the other hand, biofertilizers are applied into the microbiologicallly active soil. What is the viability of R. aquatilis JZ-GX1 in the soil inhabited by the native community of microorganisms? What is the activity of R. aquatilis JZ-GX1 in the microbiologically active soil?

Response 3: The reason why we sterilize the soil is to eliminate the interference of other microorganisms and ensure that the growth of plants is only caused by JZ-GX1 strain. As for the doubts of the reviewers, we have carried out field experiments and investigated that JZ-GX1 can also significantly promote the growth of plants including Carya illinoensis, grapes, pears and poplars in the presence of other microorganisms in the field. So JZ-GX1 strain is a good plant growth promoting bacteria.

Point 4: In the abstract is written “…colonized on the root surface, root hair and the root interior of maize.”, and in chapter 3.3 - “… JZ-GX1 is epiphytic, at least in corn.” What is truth?  The same form should be used: “indole acetic acid (IAA)” or “indoleacetic acid (IAA)”

Response 4: We are very sorry for our negligence. The cells of JZ-GX1 strain were observed on the surface, root hair and root of maize, so we thought that JZ-GX1 strain could be endophytic on maize, we have replaced “epiphytic” with “endophytic”. The revised details can be found in line 248, page 7. According to the suggestion of the reviewer, we use the expression of indole acetic acid and unify it in the full text.

Point 5: Line 11: “the growth-promoting mechanism of Rahnella aquatilis JZ-GX1” change to “the plant growth-promoting mechanism by Rahnella aquatilis JZ-GX1”

Response 5: According to the suggestion of the reviewer, we have made corrections and the revised details can be found in line 11, page 1.

Point 6: Lines 12-13: “the colonization of R. aquatilis JZGX1 in maize root” change to “the colonization of maize root by R. aquatilis JZGX1”

Response 6: According to the suggestion of the reviewer, we have made corrections and the revised details can be found in line 12-13, page 1.

Point 7: Line 65: “high phytase” change to „ high phytase activity”

Response 7: According to the suggestion of the reviewer, we have made corrections and the revised details can be found in line 66, page 2.

Point 8: Line 80: “resuspended in physiological” change to „resuspended in physiological saline solution”

Response 8: According to the suggestion of the reviewer, we have made corrections and the revised details can be found in line 81-82, page 2.

Point 9: Line 142: “pH instrument” change to „pH-meters”

Response 9: Thanks to the reviewer for your suggestions, we have made corrections and the revised details can be found in line 143, page 3.

Point 10: Line 147: “consisted of 400 g” change to “contained of 400 g”

Response 10: Thanks to the reviewer for your suggestions, we have made corrections and the revised details can be found in line 148, page 4.

Point 11: Line 159: “the response variables were determined” change to “the various parameters were determined”

Response 11: Thanks to the reviewer for your suggestions, we have made corrections and the revised details can be found in line 160, page 4.

Point 12: “measured” - Why underlined? distilledwater” - Why underlined?

Response 12: We are very sorry for our negligence. The corresponding underline has been modified, the details can be found in Line 176 and 186, Page 4.

Point 12: All figures are correctly cited in the text. The figures are informative. Table 1 is redundant. In my opinion, marking the changes significance by using letters is sufficient.

Response 12: According to the suggestion of the reviewer, we have deleted Table 1.

Round 2

Reviewer 1 Report

Dear authors,

Thank you very much for your answer, however, the identification of this microorganism shows a relevant problem. I understand that you named R. aquatilis in other manuscripts, but nowadays, the evidence says that is not R. aquatilis, maybe if you provide an accession number or some results about ANI o dDDH of its genome against other genomes from this genera could help to clarify it.

Best regards

Author Response

However, the identification of this microorganism shows a relevant problem. I understand that you named R. aquatilis in other manuscripts, but nowadays, the evidence says that is not R. aquatilis, maybe if you provide an accession number or some results about ANI o dDDH of its genome against other genomes from this genera could help to clarify it.

Reply:Dear reviewer, we have submitted the genome sequence of strain JZ-GX1 to NCBI with the accession number PRJNA720502 and we hope to clarify this matter.

Reviewer 2 Report

Dear Authors,

Thank you very much for taking into account my comments and for clarifying doubts. In my opinion, very useful information is contained in responses to the reviewer's remarks.

Research results showed, that JZ-GX1 strain can colonize on the surface and interior of maize’s roots, and therefore, phytases are present in roots. Phytases accumulated in the tissues as result of microbial activity, flow out of the roots after they have been immersed into a buffer. Although, phytases were in the roots, they were bacterial origin. The cells of this endophytic bacterium secreted phytases, and after that this enzyme could be slowly translocated into the rhizoplane and the follow into rhizosphere. Such detailed information must be included in the discussion, definitely. I propose to insert explanation of this phenomenon in line 378, in place of the sentence “This result was attributed to  bacterial colonization on the roots.”  Only then this manuscript can be published.

Additionally:

1. Lines 136 – 145 in new manuscript version.

I think that the first paragraph in chapter 2.5 can be limited to information " Soil from a mountain near Nanjing Forestry University, China, was selected because of its naturally low total P content and low level of organic matter. The soil pH was 5.1, organic matter 1.5%, and nutrient content (mg/kg): total P 3.12, effective P 0.39, K 499.92, 144 Ca 202.62, Mg 249.9, Fe 1410.6, Mn 30.66, Zn 0.24, and Cu 2.22.  These routine soil analysis was performed in the Soil Testing Laboratory at Nanjing Forestry University.”

2. Line 248 in new manuscript version.

“endogenous” should to be change to „endophytic”

3. Lines: 182, 290, 294 and 375 in new manuscript version.

According to me the term “root-secreted phytase activity” should be change”. The word “root-secreted” suggests the plant origin.

4. In the methods – subchapter 2.8 – The Authors should write what kind of phytases they have studied - emphasize that the experiment designed in this way allowed to analyze the activity of phytases derived from bacteria inhabiting the maize roots. It is worth comparing the results of phytases activity in inoculated and non-inoculated roots. Only then this manuscript can be published.

5. Line 201: delete underline of the word "distilledwater" 

Author Response

Point 1: Such detailed information must be included in the discussion, definitely. I propose to insert explanation of this phenomenon in line 378, in place of the sentence “This result was attributed to bacterial colonization on the roots.”  Only then this manuscript can be published.

Response 1: Thank you very much for the suggestions of the reviewers, which makes it easier to deepen the understanding of the article. We have rewritten it as suggested by the reviewer and the revised details can be found in line 381-387, page 13.

Point 2: I think that the first paragraph in chapter 2.5 can be limited to information " Soil from a mountain near Nanjing Forestry University, China, was selected because of its naturally low total P content and low level of organic matter. The soil pH was 5.1, organic matter 1.5%, and nutrient content (mg/kg): total P 3.12, effective P 0.39, K 499.92, 144 Ca 202.62, Mg 249.9, Fe 1410.6, Mn 30.66, Zn 0.24, and Cu 2.22. These routine soil analysis was performed in the Soil Testing Laboratory at Nanjing Forestry University.”

Response 2: According to the suggestion of the reviewer, we have rewritten this paragraph and the revised details can be found in line 138-142, page 3.

Point 3: “endogenous” should to be change to „endophytic”

Response 3: According to the suggestion of the reviewer, we have replaced the word and the revised details can be found in line 251, page 7.

Point 4: According to me the term “root-secreted phytase activity” should be change”. The word “root-secreted” suggests the plant origin.

Response 4: According to the suggestion of the reviewer, we have replaced the usage and the revised details can be found in line 185 page 4; line 294, 298 page 10; line 379 page 13.

Point 5: In the methods – subchapter 2.8 – The Authors should write what kind of phytases they have studied - emphasize that the experiment designed in this way allowed to analyze the activity of phytases derived from bacteria inhabiting the maize roots. It is worth comparing the results of phytases activity in inoculated and non-inoculated roots. Only then this manuscript can be published.

Response 5: We fully understand the concerns of the reviewer. In this study, the determination of phytase activity refers to the phytase secreted by Candida krusei, which is identified as myoinositol 2-monophosphate, so it is the phytase activity from microbial sources (See Ref. 36 for details). We have added relevant information to the materials and methods and the revised details can be found in line 186 page 4.

Point 6: Line 201: delete underline of the word "distilledwater" 

Response 6: We are very sorry for our negligence and the revised details can be found in line 204 page 5.